# Potential SARS-CoV-2 Susceptibility of Cetaceans Stranded along the Italian Coastline

**DOI:** 10.3390/pathogens11101096

**Published:** 2022-09-25

**Authors:** Tania Audino, Elena Berrone, Carla Grattarola, Federica Giorda, Virginia Mattioda, Walter Martelli, Antonio Pintore, Giuliana Terracciano, Cristiano Cocumelli, Giuseppe Lucifora, Fabio Di Nocera, Gabriella Di Francesco, Ludovica Di Renzo, Silva Rubini, Stefano Gavaudan, Anna Toffan, Roberto Puleio, Dashzeveg Bold, Francesco Brunelli, Maria Goria, Antonio Petrella, Maria Caramelli, Cristiano Corona, Sandro Mazzariol, Juergen A. Richt, Giovanni Di Guardo, Cristina Casalone

**Affiliations:** 1Istituto Zooprofilattico Sperimentale del Piemonte, Liguria e Valle d’Aosta, 10154 Torino, Italy; 2Istituto Zooprofilattico Sperimentale della Sardegna, 07100 Sassari, Italy; 3Istituto Zooprofilattico del Lazio e della Toscana, 56123 Roma, Italy; 4Istituto Zooprofilattico Sperimentale del Mezzogiorno, Portici, 80055 Napoli, Italy; 5Istituto Zooprofilattico Sperimentale dell’Abruzzo e del Molise, 64100 Teramo, Italy; 6Istituto Zooprofilattico Sperimentale della Lombardia e dell’Emilia-Romagna, 44124 Ferrara, Italy; 7Istituto Zooprofilattico Sperimentale dell’Umbria e delle Marche, 60131 Ancona, Italy; 8Istituto Zooprofilattico Sperimentale delle Venezie, 35020 Legnaro, Italy; 9Istituto Zooprofilattico Sperimentale della, Sicilia, 90129 Palermo, Italy; 10Department of Diagnostic Medicine/Pathobiology, College of Veterinary Medicine, Kansas State University, Manhattan, KS 66506, USA; 11Istituto Zooprofilattico Sperimentale della Puglia e della Basilicata, 71121 Foggia, Italy; 12Department of Comparative Biomedicine and Food Science, University of Padova, 35020 Padova, Italy; 13General Pathology and Veterinary Pathophysiology, Veterinary Medical Faculty, University of Teramo, 64100 Teramo, Italy; 14Scientific and Technical Office of REMESA Istituto Zooprofilattico Sperimentale della Sicilia, 90129 Palermo, Italy

**Keywords:** SARS-CoV-2, ACE2, CD68, marine mammals

## Abstract

Due to marine mammals’ demonstrated susceptibility to SARS-CoV-2, based upon the homology level of their angiotensin-converting enzyme 2 (ACE2) viral receptor with the human one, alongside the global SARS-CoV-2 occurrence and fecal contamination of the river and marine ecosystems, SARS-CoV-2 infection may be plausibly expected to occur also in cetaceans, with special emphasis on inshore species like bottlenose dolphins (*Tursiops truncatus*). Moreover, based on immune and inflammatory responses to SARS-CoV-2 infection in humans, macrophages could also play an important role in antiviral defense mechanisms. In order to provide a more in-depth insight into SARS-CoV-2 susceptibility in marine mammals, we evaluated the presence of SARS-CoV-2 and the expression of ACE2 and the pan-macrophage marker CD68. Aliquots of tissue samples, belonging to cetaceans stranded along the Italian coastline during 2020-2021, were collected for SARS-CoV-2 analysis by real-time PCR (RT-PCRT) (N = 43) and Immunohistochemistry (IHC) (N = 59); thirty-two aliquots of pulmonary tissue sample (N = 17 *Tursiops truncatus*, N = 15 *Stenella coeruleoalba*) available at the Mediterranean Marine Mammal Tissue Bank (MMMTB) of the University of Padua (Legnaro, Padua, Italy) were analyzed to investigate ACE2 expression by IHC. In addition, ACE2 and CD68 were also investigated by Double-Labeling Immunofluorescence (IF) Confocal Laser Microscopy. No SARS-CoV-2 positivity was found in samples analyzed for the survey while ACE2 protein was detected in the lower respiratory tract albeit heterogeneously for age, gender/sex, and species, suggesting that ACE2 expression can vary between different lung regions and among individuals. Finally, double IF analysis showed elevated colocalization of ACE2 and CD68 in macrophages only when an evident inflammatory reaction was present, such as in human SARS-CoV-2 infection.

## 1. Introduction

Coronavirus disease 2019 (COVID-19) caused by severe acute respiratory syndrome coronavirus 2 (SARS-CoV-2) has led to a global public health crisis since late 2019 [1]. Coronaviruses (CoVs) are responsible for respiratory and intestinal infections in animals and humans [2]. In addition, pneumonia impacts marine mammal health and conservation [3]. Viral pneumonia and respiratory infection have been reported as the cause of death in aquatic mammal species and populations worldwide. For example, the cetacean morbillivirus (CeMV) has caused dramatic mass die-offs of free-ranging cetaceans and unusual mortality events (UMEs) along the Italian coast during the last three decades [4].

The transmission of viral pathogens from humans to animals (reverse zoonoses) and from humans to marine mammalsincluding cetaceanshas been reported in aquatic species considered potentially susceptible to SARS-CoV-2 infection based on the homology level of their ACE2 (angiotensin-converting enzyme 2) viral receptor and the human one [5,6]. To date, evidence of SARS-CoV-2 infection has not been reported under natural conditions in any aquatic mammalian species. The survival of the virus in the aquatic environment depends in fact on several factors, such as temperature, humidity, organic matter, water type, chemicals, UVs, the presence of other organisms [7]. However, contaminated wastewater entering natural water systems could serve as a vehicle for SARS-CoV-2 transmission to susceptible pinniped and cetacean species and populations [7,8].

Preventing human-to-wildlife SARS-CoV-2 transmission is central to protect these animals from disease, some of which are classified as threatened on the International Union for Conservation of Nature [IUCN] Red list of Threatened Species (https://www.iucnredlist.org; accessed on 12 August 2022), and to impede the establishment of novel SARS-CoV-2 reservoirs in wild mammals. The risk of repeated re-infection of humans from wildlife reservoirs could severely hamper SARS-CoV-2 control efforts [5]. Marine mammals are regarded as a natural reservoir for potential zoonotic pathogens [9]. Studies on the presence of CoVs in aquatic organisms and in the marine environment [10] so far have detected one alpha and two gamma coronaviruses in marine mammals [11]. These points underscore the importance of surveillance for SARS-CoV-2 infection in the panel of *postmortem* investigations on stranded cetaceans. In Italy, this approach is taken by the network of Istituti Zooprofilattici Sperimentali, public laboratories coordinated by the Ministry of Health, that performs systematic diagnostic surveys of stranded cetaceans found dead. In the course of the current pandemic, a monitoring system has been set up *ad hoc* for the early identification of SARS-CoV-2 infection by means of molecular and immunohistochemical analysis.

Studies are needed to investigate pathogenic mechanisms underlying infection in marine mammals, including the role played by the ACE2 receptor in the spread of infection. ACE2 receptor expression was identified by immunohistochemistry (IHC) in lung tissue from cetacean species, which provided preliminary characterization of ACE2 expression in marine mammal respiratory tract [6].

Moreover, as demonstrated in humans, clinical symptoms of SARS-CoV-2 infection vary among individuals. In humans, clinical features of the infection differ between children and adults [12] possibly due to the differential expression of ACE2, the main host functional receptor for SARS-CoV-2; but data are still limited and conflicting [13]. A recent report suggests that ACE2 expression can vary by lung region and in individuals of different species [6].

Finally, the presence of ACE2 expression on macrophages’ surface has been reported. Recently, it has been also described that ACE2 expression and/or polymorphism could influence both the susceptibility to SARS-CoV-2 infection and the outcome of the associated COVID-19 disease [14]. In particular, the difference in ACE2 expression rn macrophages supports the hypothesis that lung macrophages may serve as a “Trojan horse” in antiviral defense against SARS-CoV-2 infection, thus enabling viral replication within the pulmonary parenchyma [6,15]. To test the validity of this hypothesis in marine mammals, we investigated ACE2 expression by IHC and the presence of SARS-CoV-2 in a range of body tissues of cetaceans found dead and stranded on the Italian coast between 2020–2022.

## 2. Results

### 2.1. Detection of SARS-CoV-2 in Marine Mammals Stranded along Italian Coastline

#### 2.1.1. SARS-CoV-2 RT-PCR

All samples tested (N = 61 tissue aliquots of lung, intestine, Central Nervous System-CNS, and N = 25 swabs) were negative.

#### 2.1.2. SARS-CoV-2 Immunohistochemistry

All samples tested for IHC analysis (N = 59 lung tissue aliquots) were negative for presence of SARS-CoV-2 antigens using different SARS-CoV-2 specific antibodies (AB 7B7—1:50, AB 5A—1:2000, AB 3A—1:1000) (Figure 1).

### 2.2. In-Depth Study of ACE2 Receptor

#### 2.2.1. Effects of Age, Sex, and Origin (Captivity vs. Wild) on ACE2 Expression in Cetacean Lung Tissues

IHC was performed on 32 lung tissue samples from *T. truncatus* and *S. coeruleoalba* of different ages and gender available in the Mediterranean Marine Mammal Tissue Bank [MMMTB], University of Padua, and analyzed for ACE-2 receptor expression (Table 1, Figure 2 and Figure 3). Chi-square tests to evaluate the effect of species, age, and sex suggested no statistically significant association (*p* = 0.082) between ACE-2 receptor expression and these variables, although the species variable approximated the significance level of 0.05. In contrast, although slightly above this level (*p* = 0.054), the origin of the specimen suggested an effect on ACE-2 receptor expression depending on whether the animals were of wild or captive origin.

Logistic regression using origin and species as covariates revealed no statistically significant association between ACE2 protein expression and either covariate (Table 2).

#### 2.2.2. Immunofluorescence (IF)

Lung tissue sections of four animals (Table 3, ID 123, 133, 142, 343) underwent immunofluorescence staining for ACE2 (SARS-CoV-2 receptor) and CD68 (a marker for macrophages) to determine whether the SARS-CoV-2 receptor was expressed by the cells representingg the inflammatory infiltrates seen in the lung tissue. Most epithelial cells and a subset of macrophages expressed varying degrees of ACE2 (Figure 4). Pearson’s correlation coefficient indicated significant colocalization between ACE2 and CD68 only in animal ID 123 and in ID 142 (Table 3; Figure 4C,I), where various infectious agents were identified, as previously described in the literature [16,17]. Negative controls did not express the target antigen (Appendix A).

#### 2.2.3. Western Blot

Western blot confirmed the specificity of ACE2 and CD68 primary antibodies in *S. coeruleoalba* and *T. truncatus* lung samples (Figure 5). A band was present at the expected molecular weight: 100–110 kDa for ACE2 (Figure 5A,C) and 60 kDa for CD68 (Figure 5B,D).

## 3. Discussion

Transmission of human pathogens to non-human animals, including wildlife, occurs more regularly than often thought [18,19]. SARS-CoV-2 appears to have a striking ability to infect a broad range of distantly related mammals. Due to its high transmissibility and prevalence, the virus may spread to susceptible, wild, non-human mammal populations. As we enter the third year of the pandemic, SARS-CoV-2 infection continues its global spread. Multiple factors are driving its transmission. As the virus continues to evolve, the emergence of variants poses new challenges to public health.

SARS-CoV-2 RNA has been demonstrated in the wastewater and rivers of countries with high COVID-19 caseloads [20,21]. The coastal ocean is the ultimate sink for urban sewage. The risk of SARS-CoV-2 transmission in the receiving coastal water bodies should not be underestimated, though its stability in water is lower than that of other known non-enveloped human enteric viruses with waterborne transmission [20,22]. Regarding the marine environment, SARS-CoV-2 will be exposed to an aggressive assault because of UV radiation and heat; salinity and PH negatively affect viral vitality and viability in the marine environment; by contrast, plastic and organic material influence positively the viral persistence in the environment [23]. Virus concentration reduces rapidly at high temperatures but can persist long in cold waters. At high titer it may survive more than 20 days at 4 °C and for 7 days at 22 °C [7].

Viral concentration decreases rapidly at high temperatures but its survival in cold waters remains a threat for marine mammals.

Viruses in raw wastewater are not readily removed by treatment and thus become environmental pollutants. Although the ocean provides for rapid dilution of sewage, its self-depuration capacity is finite, especially in coastal areas. Marine water may become a conduit for zoonotic transmission of SARS-CoV-2 to marine wildlife at a virus concentration above the infection dose level. The range of infectious doses for direct seawater contact is unknown and the degree of exposure is difficult to estimate; nonetheless, a potential impact assessment of virus pollution in coastal marine waters is warranted.

In addition, as the Mediterranean is a “closed sea basin”, a “concentrating activity” towards chemical pollutants as well as towards infectious pathogens is possible.

In fact, microplastic pollution is one of the emerging threats across the globe and is becoming a topic of intense study for environmental researchers and the Mediterranean Sea has been recognized as a target hotspot of the world [24].

As recently described in the literature, pathogens are capable of associating with microplastics in contaminated seawater, with more parasites adhering to microfiber surfaces as compared with microbeads. Given the global presence of microplastic, this could be a novel pathway by which anthropogenic pollutants may be mediating pathogen transmission in the marine environment, with important ramifications for wildlife and human health [25].

In the ongoing COVID-19 pandemic, direct and indirect exposure occurs when animals share the same space with infected humans. Positive molecular findings in animals cohabiting with COVID-19-positive humans (e.g., pets, farmed mink, big cats, and gorillas in zoos [23]), have raised concern about the role animals may play in amplifying and spreading the virus and establishing reservoirs in the vicinity of humans. Animal infection studies published so far suggest that SARS-CoV-2 efficiently replicates in ferrets, cats [26], and rabbits [27] but poorly in dogs, pigs [26], and cattle [28]. Genome analysis strongly suggests that zoonotic spillover of SARS-CoV-2 from farmed mink to humans occurred in the Netherlands and Poland [29,30]. Researchers at the Centro de Pesquisa e Conservação de Mamíferos Aquáticos do Instituto Chico Mendes de Biodiversidade (ICMBio/CMA) in Brazil found that Antillean manatees can be infected with SARS-CoV-2 and confirmed the first case of COVID-19 in a *Sirenia* species [31].

Current knowledge indicates that wildlife does not play a significant role in the transmission of SARS-CoV-2 to humans, but its spread in animal populations can affect their health status and facilitate the emergence of new virus variants. In addition to domestic animals, free-ranging, captive, or farmed wild animals (e.g., big cats, minks, ferrets, North American white-tailed deer, and great apes) have been found positive for SARS-CoV-2 infection [21]. To date, farmed mink and pet hamsters have been shown capable of infecting humans with the SARS-CoV-2 virus. The introduction of SARS-CoV-2 to wildlife could allow the establishment of animal reservoirs.

As in terrestrial species, emerging viruses in marine mammals may be associated with other diseases such as neoplasia, epizootics, and zoonotic disease, involving a complex pathogenesis with noninfectious cofactors such as anthropogenic toxins, biotoxins, immunologic suppression, and other environmental stressors.

Surveillance and assessment for SARS-CoV-2 in marine environments should be activated to eliminate COVID-19; caution is needed to prevent SARS-CoV-2 transmission to threatened species and aquatic populations in the vicinity of human activities. The Food and Agriculture Organization (FAO), the World Organization for Animal Health (OIE), and the World Health Organization (WHO) have called on countries to take necessary steps to reduce the risk of SARS-CoV-2 transmission between humans and wildlife, to reduce the risk of new variants, and to protect human and wildlife health.

To date, despite the presumed susceptibility of marine mammals to SARS-CoV-2 [1,6,7,32], infection of wild marine mammals had not yet been confirmed. Due to many cetacean species being classified as critically endangered and exposed to escalating anthropogenic stressors, we, as a National Reference Centre for Diagnostic Investigations on Stranded Marine Mammals, decided to increase the surveillance of cetaceans and screen cetaceans stranded on the Italian coast for SARS-CoV-2 infection during the COVID-19 pandemic.

Our study underlines the importance of monitoring marine mammal populations for SARS-CoV-2 infection. Through the collaboration of the network of Istituti Zooprofilattici Sperimentali, we are carrying out a monitoring activity on marine mammals stranded on the Italian coast which are known to be potentially susceptible to SARS-CoV-2. Such measures are intended to protect the health of animals and humans.

In addition, the number of animal cases of SARS-CoV-2 infection, albeit often occasional, continues to rise. Within this context, white-tailed deer in the United State (Eastern and Midwest States) have attracted attention after several human SARS-CoV-2 “variants of concern” and “variants of interest” were identified in the deer. According to the OIE Report “SARS-CoV-2 in Animals” from 30.04.2022, 675 outbreaks in animals have been reported globally, affecting 23 species in 36 countries (https://www.woah.org/app/uploads/2022/05/sars-cov-2-situation-report-12.pdf; accessed on: 30 April 2022).

With a virus so competent at spilling over into a range of phylogenetically unrelated species, the fear is that—though the pandemic is presently under control in human populations—the virus could remain in an animal population (especially in the wild), ready to spill back into humans and trigger a new epidemic cycle. In this regard, the suspected origin of SARS-CoV-2 from a primary animal reservoir (most likely a *Rinolophus* spp. bat), followed or not by an intermediate (and hitherto unknown) host species, should be kept in mind, because as in other agents responsible for “emerging infectious diseases”, the proven or suspect origin is the animal kingdom in at least 70% of them [33].

Another aim of the present study was to evaluate the expression and the effect of age, sex, and species on the expression of the viral host cell receptor ACE2 in the lung tissue of *S. coeruleoalba* and *T. truncatus*. Previous studies investigating ACE2 receptor expression in animal species hypothesized the susceptibility of some species to SARS-CoV-2 [7,32]. Univariate analysis of ACE2 receptor expression and origin of cetacean specimens (*p* = 0.054) suggested an effect on receptor expression. To assess the direction of this effect, we performed regression analysis but found no statistically significant association for ACE2 protein expression, although captive individuals appeared to express ACE2 more often than wild individuals; the small study sample size might explain the lack of statistical significance despite this apparent difference. Further investigation on more tissue samples is therefore warranted.

Neutrophil granulocytes and macrophages play a key role in inflammatory and immune response in terrestrial and aquatic mammals. We noted in the cetacean lung tissues a peculiar mode of compartmentalization of macrophages that suggested different functional specialization. The macrophage cytotype, the so-called pulmonary intravascular macrophage (PIM), probably involved in the uptake and subsequent phagocytosis of foreign elements (particulate matter of physical or biological origin) is carried in the blood. It plays a role complementary to the other macrophage cytotype residing in the lung, namely, the alveolar macrophage [34]. As revealed by IF, ACE2/CD68 co-localization was higher when an infectious agent was present and was probably on activated type 1 macrophages (M1). These findings indicate that in cetaceans alveolar macrophages and PIM, as frontline immune cells, may be directly targeted in the course of SARS-CoV-2 infection [35].

Statistical analysis showed a significant difference in ACE2 receptor expression between wild and captive individuals. Although this finding needs further analysis in a larger sample, one possible explanation is the level of chronic stress to which captive specimens are exposed, which could lead to higher cortisol production in these animals. Given the anti-inflammatory action of cortisol, this could have implications for the production of ACE2 (which has an antiphlogistic action). Moreover, the frequent use of drugs in captive animals including anti-inflammatory drugs could be a valuable explanation.

The differential susceptibility of domestic and wild animal species to SARS-CoV-2 infection is primarily driven by their homology with human ACE2 and the viral receptor’s region directly interacting with the viral spike (S) protein receptor-binding domain (RBD). The homology with other SARS-CoV-2 host cell receptors, such as neuropilin 1 (NP1) [36] could be an area of focus for further studies.

## 4. Materials and Methods

### 4.1. Sampling

A complete *postmortem* examination was performed on stranded specimens during the years 2020–2022 according to standardized protocols [37]; during the necroscopic investigations an *ad hoc* protocol for the execution of swabs on site has been prepared (data not shown).

### 4.2. SARS-CoV-2 Real Time-PCR (RT-PCR)

RT-PCR analysis was carried out *ad hoc* on 43 cetaceans (n = 29 *S. coeruleoalba*, n = 12 *T. truncatus*, n = 1 *Z. cavirostris*, n = 1 *G. griseus,*
Table 4*)* stranded on the Italian coast during 2020–2021; tissue samples of lung, intestines, and central nervous system (CNS) and swabs (oropharyngeal, tracheal, blowhole, rectal) were collected for analysis (Table 4).

### 4.3. RNA Extraction and SARS-CoV-2 RT-PCR Detection

Total RNA was extracted from swabs as described below: 200 μL of swab medium (COPAN Diagnostics Inc., Brescia Italy) were processed in an automated extractor (MagMax Viral Isolation kit for KingFisher, Thermo Fisher, Waltham USA, or Maxwell^®^ RSC Viral Total Nucleic Acid Purification Kit for Maxwell RSC, PROMEGA, Milano Italy) according to the manufacturer’s instructions.

Ten μL of eluted RNA were then used for RT-qPCR assay using commercial kits: GeneFinder COVID-19 Plus Real Amp Kit (EliTechGroup, Puteaux France), Taqman 2019-nCov assay kit v2 (Thermo Fisher), Elitec GeneFinder COVID-19 Plus Real Amp Kit (EliTechGroup). Amplification on a QuantStudio™ 5 Real-Time PCR System (Thermo Fisher). Results are based on multiplex amplification of three different SARS-CoV-2 genomic regions [Orf1ab, N (nucleocapsid) gene, and S (spike protein) gene], over the internal process control with an MS2 phage.

### 4.4. SARS-CoV 2 Immunohistochemistry

One section of a lung tissue sample from 59 cetaceans (n = 42 *S. coeureoalba*, n = 14 *T. truncatus*, n = 1 *Z. cavirostris*, n = 1 *G. griseus*, n *= 1 B. physalus*, Table 4-Figure 6-Appendix A) found stranded and dead on the Italian coast during 2020-2022 was collected for SARS-CoV-2 analysis by IHC to determine the presence of SARS-CoV-2. Specimens ID 1 to 43 also underwent molecular analysis; specimens ID 44 to 59 underwent only IHC. 

IHC staining for SARS-CoV-2 was performed. Lung tissue sections were formalin-fixed and paraffin FFPE, then sectioned and hydrated with xylene; endogenous peroxidase was blocked using a 3% hydrogen peroxide solution. Heat-induced antigen retrieval was performed using a citrate buffer bath (pH 6.1) at 97 °C for 15 min. After cooling to room temperature, the sections were incubated with blocking serum (Vectastain ABC Kit Mouse IgG for 7B7 (Fisher Scientific, USA), Vectastain ABC kit Peroxidase Rabbit IgG for 5A and 3A) for 20 min before being incubated overnight with the primary antibody (7B7, a murine monoclonal anti-RBD protein antibody, diluted 1:50; 5A, an anti-S protein rabbit polyclonal antibody, diluted 1:2000; 3A, an anti-N protein rabbit polyclonal antibody, diluted 1:1000 kindly donated by Prof. Juergen A. Richt (Kansas State University, College of Veterinary Medicine, Waltham, MA, USA) at 4 °C in tris-buffered saline (TBS) containing Tween (Table 5). The next day, after washing with TBS buffer, the sections were incubated with the secondary antibody for 30 min, then with the prepared Vectastain ABC reagent; final detection was carried out with diaminobenzidine (DAB, Dako K3468, Abcam, Cambridge, UK) as chromogen for 10 min. The sections were counterstained with Mayer’s hematoxylin for better visualization of tissue morphology. Adequate positive and blank control tissues were utilized in each run.

**Table 4 pathogens-11-01096-t004:** Samples tested for SARS-CoV 2 by RT-PCR and IHC.

Year	ID (Case PCR/IHC)	Organ	Swab	Species
2020	1	Lung		*T. truncatus*
	2	Lung		*S. coeruleoalba*
	3	Lung		*S. coeruleoalba*
	4	Lung		*S. coeruleoalba*
	5	Lung		*Z. cavirostris*
	6	Lung		*S. coeruleoalba*
	7	Lung		*S. coeruleoalba*
	8	Lung		*S. coeruleoalba*
	9	Lung		*S. coeruleoalba*
	10	Lung		*T. truncatus*
	11	Lung		*S. coeruleoalba*
	12	Lung		*T. truncatus*
	13	Lung		*T. truncatus*
	14	Lung		*S. coeruleoalba*
	15	Lung		*S. coeruleoalba*
	16	Lung		*G. griseus*
	17	Lung		*T. truncatus*
	18	Lung		*S. coeruleoalba*
	19	Lung		*T. truncatus*
	20	Lung		*S. coeruleoalba*
	21	Lung		*S. coeruleoalba*
	22	Lung		*S. coeruleoalba*
	23	Lung		*S. coeruleoalba*
	24	Lung		*S. coeruleoalba*
	44 *	Lung		*S. coeruleoalba*
	45 *	Lung		*S. coeruleoalba*
	46 *	Lung		*S. coeruleoalba*
	47 *	Lung		*S. coeruleoalba*
	48 *	Lung		*S. coeruleoalba*
	49 *	Lung		*S. coeruleoalba*
	50 *	Lung		*S. coeruleoalba*
	51 *	Lung		*S. coeruleoalba*
	52 *	Lung		*S. coeruleoalba*
	53 *	Lung		*S. coeruleoalba*
	54 *	Lung		*S. coeruleoalba*
	55 *	Lung		*T. truncatus*
	56 *	Lung		*Balaenoptera physalus*
	57 *	Lung		*S. coeruleoalba*
	58 *	Lung		*S. coeruleoalba*
	59 *	Lung		*T. truncatus*
2021	25	Lung		*S. coeruleoalba*
		Intestine		
		CNS		
	26	Lung	Rectum	*S. coeruleoalba*
		Intestine		
		CNS		
	27	Lung		*S. coeruleoalba*
		CNS		
	28	Lung	Oropharynx	*S. coeruleoalba*
		Intestine	Rectum	
		CNS	Trachea	
			Blowhole	
	29	Lung		*S. coeruleoalba*
	30	Lung		*S. coeruleoalba*
	31	Lung		*S. coeruleoalba*
	32	Lung		*S. coeruleoalba*
	33	Lung		*S. coeruleoalba*
	34	Lung		*S. coeruleoalba*
	35	Lung		*S. coeruleoalba*
	36	Lung		*S. coeruleoalba*
	37	Lung	Oropharynx	*T. truncatus*
		Intestine	Rectum	
		CNS	Trachea	
			Blowhole	
	38	Lung		*T. truncatus*
		CNS		
	39	Lung	Oropharynx	*T. truncatus*
		Intestine	Rectum	
		CNS	Trachea	
			Blowhole	
	40	Intestine		*S. coeruleoalba*
	41	Lung	Oropharynx	*T. truncatus*
		Intestine	Rectum	
		CNS	Trachea	
			Blowhole	
2022	42	Lung	Oropharynx	*T. truncatus*
		Intestine	Rectum	
		CNS	Trachea	
			Blowhole	
	43	Lung	Oropharynx	*T. truncatus*
		Intestine	Rectum	
		CNS	Trachea	
			Blowhole	

Legend: CNS Central Nervous System; ID (case PCR/IHC) reference number of specimen; ID (case PCR/ IHC) reference number of lung specimens analyzed by IHC and PCR; * specimens that underwent only IHC.

### 4.5. Immunofluorescence

Drawing on the inflammatory and immune response to SARS-CoV-2 infection in humans, we investigated the expression of ACE-2 and CD68 in macrophages by means of double immunofluorescence staining. To do this, selected formalin-fixed, paraffin-embedded (FFPE) lung tissue sections 4 ± 2 μm thick (Table 2 and Table 3 ) were processed for IF analysis.

In detail, antigen retrieval was performed using 10 mM citrate buffer (pH 6.1) at 95 °C for 20 min. The sections were incubated in a blocking buffer (5% normal donkey serum, 0.3% Triton X-100 in 0.01 M PBS, pH 7.4) for 1 h at room temperature, then incubated for 24–48 h at 4 °C in a solution of 0.01 M PBS, pH 7.4, containing 0.1% Triton X-100, 2% normal donkey serum, and the primary antibodies. Commercially available primary antibodies (Abs) were used: an anti-ACE2 polyclonal antibody (1:200 overnight, Abcam#ab15348) and an anti-CD68 monoclonal antibody (1:50 overnight, DAKO#M0718) (Table 5). After several washes, the sections were incubated with appropriate solutions of donkey Alexa 488 or Alexa 555 conjugated secondary antibodies (1:1000, Thermo Fisher). The slides were then washed in PBS, counterstained with 4,6-diamidino-2-phenylindole (DAPI, 1:1000, KPL, USA), and mounted with Fluoromount G (SouthernBiotech, USA). As negative internal controls, primary antibodies were eliminated and replaced with nonimmune homologous serum (Appendix A). All fluorescence images were captured on a confocal laser scanning microscope (Leica TCS SP8, Leica Microsystem, Germany).

**Table 5 pathogens-11-01096-t005:** Primary antibodies in double IF and IHC.

Antigen	Target	Antibody/Antiserum	Host	Dilution	Source	Technique(s)
**7B7**	RBD protein	Mono	Mouse	1:50	Kansas State University College of Veterinary Medicine	IF, IHC
**5A**	S protein	Poly	Rabbit	1:2000	Kansas State University College of Veterinary Medicine	IHC
**3A**	N protein	Poly	Rabbit	1:1000	Kansas State University College of Veterinary Medicine	IHC
**ACE 2**	ACE2 receptor	Poly	Rabbit	1:200/1:2000	Abcam	IF, IHC
**CD 68**	Macrophage	Mono		1:50	DAKO	IF

RBD denotes receptor binding domain; N denotes nucleocapsid; S denotes spike protein; ACE2, angiotensin-converting enzyme 2; poly, polyclonal; mono, monoclonal; IF, immunofluorescence; IHC, immunohistochemistry.

### 4.6. Western Blot

ACE2 and CD68 primary antibodies were validated by means of Western blot for their use in striped dolphin brain samples. Central nervous system (CNS) specimens from *Stenella coeruleoalba* were homogenized at 10% weight/volume and subjected to immunoblotting. Twenty to forty μg of total protein for each sample were reduced, loaded on MiniProtean TGX 4–20% gels (#456–1094, Bio-Rad, USA), and separated electrophoretically in XT MES running buffer 1X (Bio-Rad) for about 50 min at 150 V using a MiniProtean II electrophoretic chamber (Bio-Rad). Protein transfer was obtained at 25 V for 5 min using a semi-dry Trans-Blot-Turbo (Bio-Rad) transfer system according to the manufacturer’s protocols. Membranes were blocked by an I-Block reagent (Thermo Fisher #T2015) for 1 h at room temperature. Detection of proteins was carried out, respectively, with anti-ACE2 polyclonal antibody (1:1000 overnight, Abcam #ab15348) or CD68 monoclonal antibody (1:500 overnight, DAKO #M0718). Membranes were incubated with a 1:10,000 dilution of HRP-goat anti-rabbit antibody (Invitrogen #656120, Thermo Fischer) or HRP-rabbit-anti-mouse Ig G (Sigma #A9044, Sigma Aldrich, USA) and with 1 mL of the combined 1:1 solution of the Clarity™ Western ECL Substrate detection kit (Bio-Rad). Finally, membranes were acquired in chemiluminescence using the ChemiDoc™ Touch (Bio-Rad) image acquisition or equivalent system.

### 4.7. ACE-2 Immunohistochemistry

Lung tissue samples obtained from the Mediterranean Marine Mammal Tissue Bank (MMMTB), University of Padua (Legnaro, Padua, Italy), were analyzed to determine a possible relationship between ACE-2 expression in lung tissue and animal age and sex (n = 17 *T. truncatus*, n = 15 *S. coeruleoalba*, Table 2-Appendix A). IHC staining for ACE-2 was performed as outlined below.

A total of 32 FFPE lung tissue from *T. truncatus* and *S. coeruleoalba* of different ages were sectioned and hydrated with xylenes; endogenous peroxidase was blocked using a 3% hydrogen peroxide solution. Heat-induced antigen retrieval was performed using a citrate buffer bath (pH 6.1) at 97 °C for 15 min. After cooling to room temperature, the sections were incubated with blocking serum (Vectastain ABC Kit, pk-4001, Fisher Scientific) for 20 min before being incubated overnight with the primary anti-ACE-2 antibody (polyclonal antibody raised in rabbits and diluted 1:2000, ab15348 Abcam, UK) at 4 °C in tris-buffered saline (TBS) containing Tween (Table 5). The next day, after washing with TBS buffer, the sections were incubated with the secondary antibody for 30 min, then with the prepared Vectastain ABC reagent; final detection was carried out with diaminobenzidine (DAB, Dako K3468, Abcam, UK) as chromogen for 5 min. The sections were then counterstained with Mayer’s hematoxylin for better visualization of tissue morphology. Adequate positive and blank control tissues were utilized in each run.

The results were expressed by means of a score (+/++/−) derived from the independent analysis carried out by 3 experienced pathologists.

### 4.8. Statistical Analysis

The map of the stranding sites (Figure 6) was produced using QGIS software version 3.10.2-A Coruña from an Excel file containing the geographical coordinates for all animals. A coordinate along the coastline close to the stranding site was used in cases in which no stranding data were available.

As regards data analysis of ACE-2 expression, given the categorical nature of the variables, the dataset of 32 cetacean samples was analyzed using Chi-square tests to assess the relationship between ACE-2 expression and the variables species, origin, age, and sex. Given the small sample size, we evaluated variables with more than two categories in a binary manner, conflating weakly (+) and strongly expressed (++) ACE2 and the “Calf” and “Juvenile” subjects in a single category for the age variable.

In order to assess the direction of the effects, logistic regression was performed using the variables considered useful as covariates based on the previous chi-square tests. A *p*-value < 0.05 (*p* < 0.05) was considered statistically significant.

Quantification of colocalization was performed using Pearson’s correlation coefficient. Each pairwise comparison was performed on 5 sets of images acquired with the same optical settings. Pearson’s correlation coefficients greater than > 0.5 were interpreted as indicative of reliable colocalization between 2 spectrally separated fluorophores [38].

## 5. Conclusions

There has been little evidence for SARS-CoV-2 in Mediterranean Sea waters; nonetheless, the risk of its transmission into coastal waterbodies merits attention. Although we found no evidence of SARS-CoV-2 spillover in cetaceans stranded on the Italian coast and monitored by our network, ACE2 expression in lung tissues suggests a potential susceptibility of marine mammals to SARS-CoV-2 infection. From a One Health perspective, monitoring stranded specimens for systematic surveillance of SARS-CoV-2 infection in marine mammals is essential to protect human health and that of endangered marine mammal species.

## Figures and Tables

**Figure 1 pathogens-11-01096-f001:**
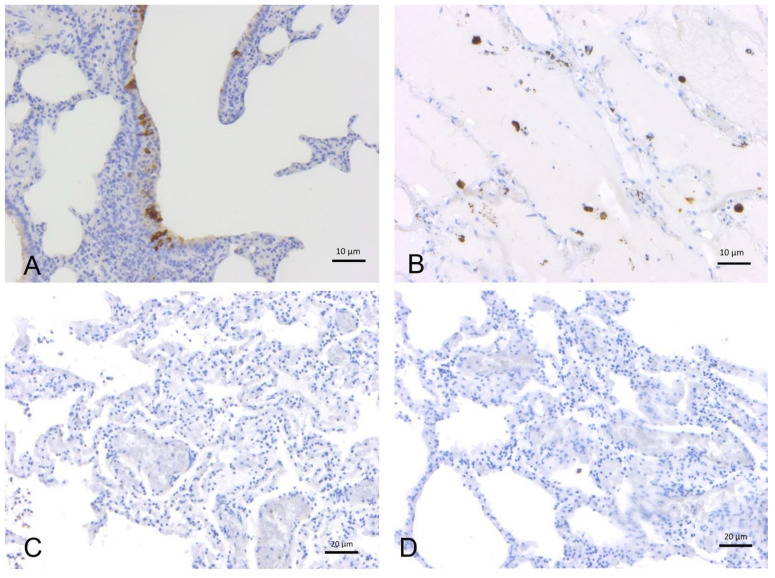
Immunohistochemical analysis with anti-SARS-CoV-2 polyclonal antibodies (3A, 5A) (**A**) Human lung (positive control): positive immunoreactivity within type I pneumocytes from alveolar respiratory epithelium, 3A polyclonal Ab. (**B**) Hamster lung (positive control): positive labeling of alveolar macrophages, 5A polyclonal Ab. (**C**) Bottlenose dolphin (*T. truncatus*) lung: absence of staining, 5A polyclonal Ab. (**D**) Bottlenose dolphin lung: absence of staining, 3A polyclonal Ab.

**Figure 2 pathogens-11-01096-f002:**
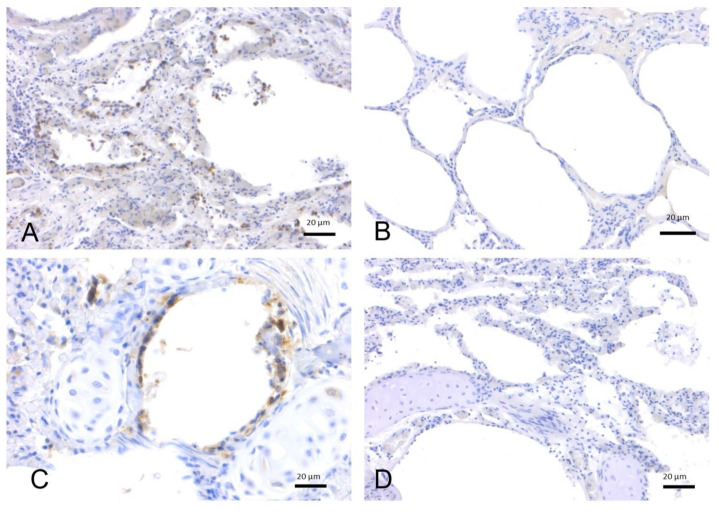
Immunohistochemical analysis by means of an anti-ACE2 polyclonal antibody on lung tissues from bottlenose dolphins (*T. truncatus*). (**A**) Lung, adult (ID 133–Appendix A). Positive labeling on the surface of alveolar (type I pneumocytes) and bronchiolar epithelial cells. (**B**) Lung, adult (ID 201–Appendix A). Absence of staining. (**C**) Lung, juvenile (ID 349–Appendix A). Type I pneumocytes and bronchiolar epithelial cells are positive for ACE2. (**D**) Lung, calf (ID 359–Appendix A). Absence of staining.

**Figure 3 pathogens-11-01096-f003:**
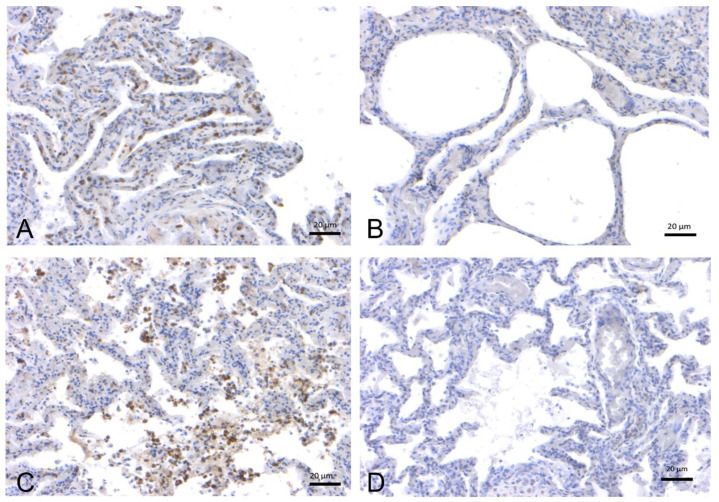
Immunohistochemical analysis by means of an anti-ACE-2 polyclonal antibody on lung tissues from striped dolphins (*S. coeruleoalba*). (**A**) Lung, adult (ID 167–Appendix A). Positive labeling in type I pneumocytes from the alveolar respiratory epithelium and in alveolar macrophages. (**B**) Lung, adult (ID 447–Appendix A). Absence of staining. (**C**) Lung, juvenile (ID 255–Appendix A). Type I pneumocytes and alveolar macrophage are positive for ACE-2. (**D**) Lung, calf (ID 374–Appendix A). Absence of staining.

**Figure 4 pathogens-11-01096-f004:**
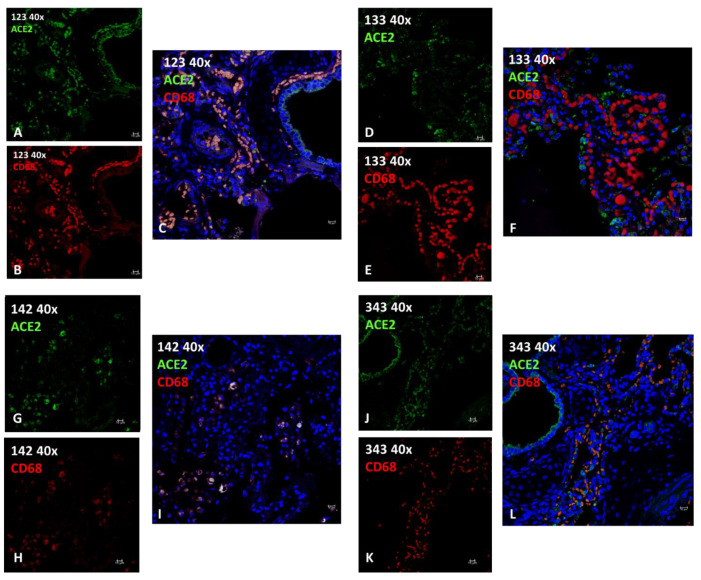
Immunofluorescence analysis of ACE2 and CD68 (macrophage) in lungs from *T. truncatus*. (**A**) ID 123 ACE2 (green) positive cells. (**B**) ID 123 CD68 (red) positive cells. (**C**) ID 123 Merge ACE2 (green) and CD68 (red) with co-localization in yellow. (**D**) ID 133 ACE2 (green) positive cells. (**E**) ID 133 CD68 (red) positive cells. (**F**) ID 133 Merge ACE2 (green) and CD68 (red) No co-localization. (**G**) ID 142 ACE2 (green) positive cells. (**H**) ID 142 CD68 (red) positive cells. (**I**) ID 142 Merge ACE2 (green) and CD68 (red) with co-localization in yellow. (**J**) ID 343 ACE2 (green) positive cells. (**K**) ID 343 CD68 (red) positive cells. (**L**) ID 343 Merge ACE2 (green) and CD68 (red) No co-localization. Blue: 4,6-diamidino-2-phenylindole (DAPI). Scale bars, 10 µm.

**Figure 5 pathogens-11-01096-f005:**
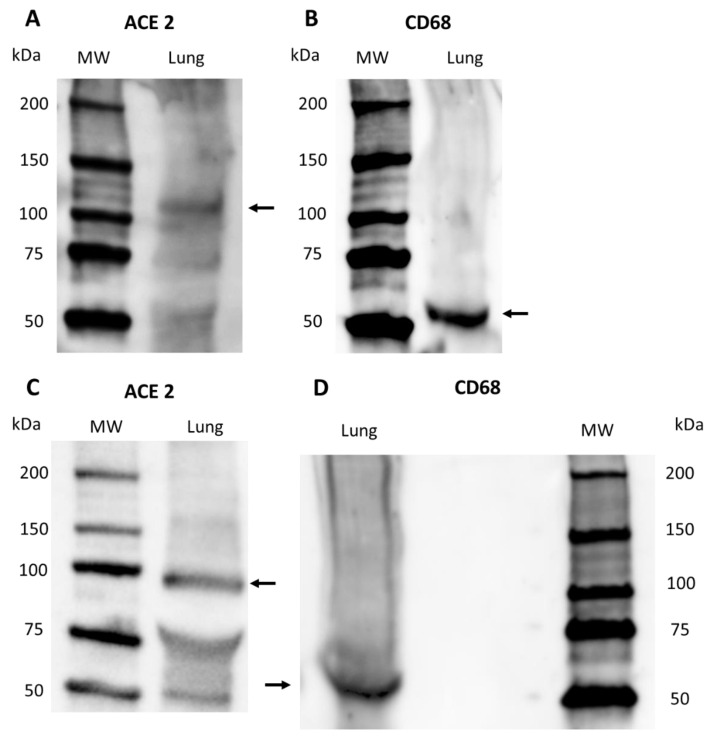
Western Blot of ACE2 and CD68 protein expression on *S. coeruleoalba* and *T. truncatus* dolphin lung samples: (**A**) ACE2 in *S. coeruleoalba* lung sample; (**B**) CD68 in *S. coeruleoalba* lung sample; (**C**) ACE2 in *T. truncatus* lung sample; (**D**) CD68 in *T. truncatus* lung sample. Arrows indicates the correct bands. MW, molecular size markers (in kilodaltons).

**Figure 6 pathogens-11-01096-f006:**
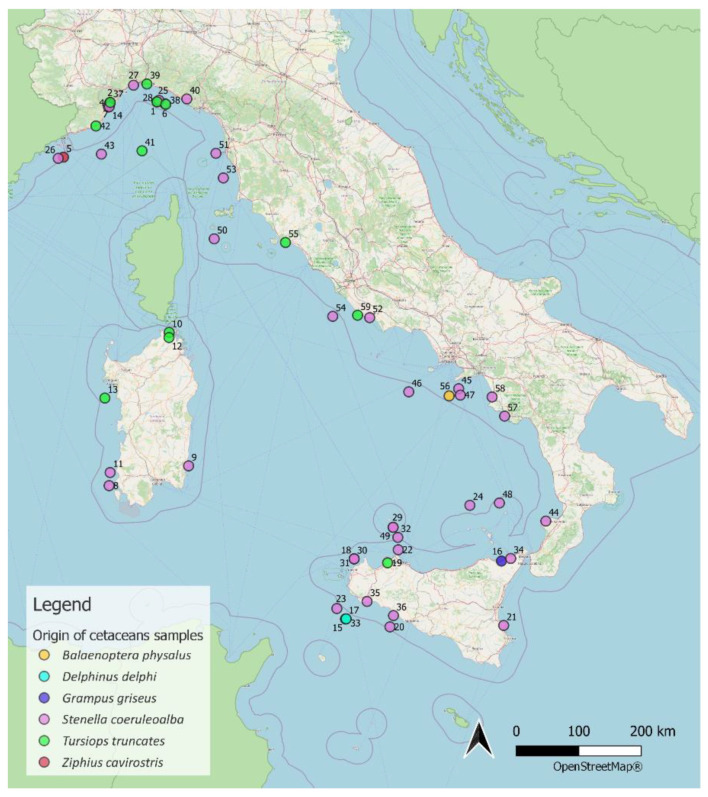
Map of stranding sites.

**Table 1 pathogens-11-01096-t001:** ACE2 immunohistochemical analysis of lung tissue from the Mediterranean Marine Mammal Tissue Bank (MMMTB), University of Padua (Legnaro, Padua, Italy).

Species	n°	Sex		Age			Origin		IHC ACE2		
		Female	Male	Juvenile	Calf	Adult	Captivity	Wild	−	+	++
*T. truncatus*	17	9	8	2	6	9	10	7	3	5	9
*S. coeruleoalba*	15	8	7	6	2	7	0	15	8	2	5

Legend of Results of ACE2 expression by IHC: ++ Highly expressing sample; + Weakly expressing sample; − Negative sample. (these scores resulted from the independent analysis carried out by 3 experienced pathologists, with the results of their evaluations/investigations being summarized through "median/average” data).

**Table 2 pathogens-11-01096-t002:** Univariate analysis and logistic regression.

	Chi-square Test for ACE-2 by IHC	Regression Analyses for ACE2 by IHC	
	*p*–value	Odds ratio	95% Conf. Interval
Species	0.082	0.525 ¹	0.091–3.034
Origin	0.054	4.8 ²	0.385–59.895
Age	1	/	/
Sex	0.647	/	/

¹ base category for comparison: *T. truncatus*. ² base category for comparison: wild.

**Table 3 pathogens-11-01096-t003:** Clinico-pathological data and results of ACE2 and CD68 expression in lung by immunofluorescence (IF).

ID	Species	Sex	Age	Origin	Cause of Death	Infections	Pathology (EE LUNG)	IF			Note
								ACE2	CD68	Colocalization (Person’s Coefficient/%)	
123	*T. truncatus*	F	Calf	C	Birth hypoxia with meconium aspiration syndrome (MAS)	*Burkholderia cepacia, Aeromonas hydrophila*	meconium in 46% of 50 fields observed at 40X; diffuse interstitial lympho-plasma cell infiltrates with edema, hemorrhage and hyperemia; hypertrophic and multifocally hyperplastic bronchial and bronchiolar mucosa; macrophages.	+++	+++	0.8580 98.35%	presence of active macrophages due to infection
133	*T. truncatus*	F	Adult	C	ND	ND	large areas of severe mixed interstitial infiltrates (lymphocytes, plasma cells, macrophagesand neutrophilic granulocytes); abundant macrophage exudation with rare neutrophilic granulocytes; both alveolar and bronchial with diffuse tissue edema; diffuse anthracosis.	++	+++	0.4137 5.19 %	Absence of active macrophages (M1) due to lack of infection
142	*T. truncatus*	F	Adult	W	Sepsis resulting from systemic mycosis and Toxoplasma gondii encephalitis	*Toxoplasma gondii*	in bronchial lumen macrophage and lymphocytic inflammation; severe exudation in some bronchioles of active macrophages and neutrophils; necrosis and fungal hyphae. peripheral vascular structures with edema and fibrinoid necrosis associated with thrombosis with fungal hyphae.; diffuse mild fibrosis of septa with macrophage activation and edema.	+	+	0.8644 89.52 %	presence of active macrophages due to infection, in this case parasitic.
343	*T. truncatus*	F	Calf	C	meconium-induced colic constipation.	ND	areas of emphysema, mainly in the sub-pleural area; large portions of parenchyma characterized by neonatal atelectasis and scarce keratin scales in the alveolar spaces; prominent intravascular pulmonary macrophages; occasional circulating neutrophilic granulocytes (neutrophilic margination); flaking erythrocytes and epithelial cells; exogenous materials are noted. minimal macrophage exudation	+ (+++ in epithelial tissue)	+++	0.5870 25.01 %	Absence of active macrophages (M1) due to lack of infection

Abbreviations: ID = identity; C = Captivity; W = Wild ND = not detected; HE = Hematoxylin Eosin; − = no expression; + = weak expression; ++ = moderate expression; +++ = strong expression.

## Data Availability

The data presented in this study are available within the article.

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
