# Peer review of "Potential SARS-CoV-2 Susceptibility of Cetaceans Stranded along the Italian Coastline"

_pathogens, 2022, doi:10.3390/pathogens11101096_

Round 1

Reviewer 1 Report

This is an important, timely manuscript that significantly advances our understanding of the susceptibility of cetaceans to SARS-Co-2, and thus warrants publication in Pathogens. The introduction is well written and explains the need for this work and thoroughly reviews previous research on the topic.

The results are clear and unequivocal. The figures are beautiful and clearly demonstrate the research findings. The statistical analyses chosen are appropriate to the data and clearly presented.

A few specific minor editorial suggestions”

Line 26, change mammal’s to mammals’

Line 29 change ecosystem to ecosystems

Line 30 change enphasis to emphasis

Line 38 change thirtytwo to thirty-two

Line 44 change specie to species

Line 55 remove on after impacts

Lines 109 and 112 Change resulted to were

Throughout change gender to sex

Author Response

Thank you for your positive feedback about our work. I have corrected the manuscript based on your editorial suggestions.

Kind regards

Cristina Casalone

Reviewer 2 Report

This paper summarizes the results of a systematic Sars-Cov-2 diagnostic survey of stranded cetaceans found dead along the Italian coast.

Following up on a previous study published last year (Audino et al., Animals, 2021), the authors screened 4 dolphin species (mostly S. coeruleoalba (n=29) and T. truncatus (n=12)) from the Italian coast for the presence of Sars-Cov-2 by RT-PCR and immunohistochemistry (IHC). Most of the analysed specimen were sampled from lungs but also intestine, central nervous systems and swabs (oropharynx, blowhole, trachea and rectum). None of the tested samples revealed positive for the presence of Sars-Cov-2 either by RT-PCR (76 samples in total) or by IHC (59 samples, 3 antibodies against Sars-Cov-2 were tested).

In an effort to better understand the susceptibility of dolphins to Sars-CoV-2, the authors analysed the expression of ACE2 in the lungs (and to a lesser extent in the brain) of 2 dolphin species by IHC and western blot. Then, they authors performed colocalization experiment between ACE2 and the macrophage receptor CD68 by IF in lungs samples of T. truncatus.

Introduction

Line 96-98: this sentence needs to be clarified.

Results

Regarding the negative results obtained by RT-qPCR, is the nucleotide sequence from Sars-CoV-2 infecting marine mammals available? If so, are the commercial kits used in this study suited to screen the dolphin samples? In other words, were the primer/probes designed in conserved regions across human and dolphin virus? One may also worry about PCR inhibitors or RNA degradation that could be present in decaying / stranded tissue samples. Did the authors use internal positive control for the RT-qPCR assay to check for the RNA integrity?

Regarding the sampling, which region of the lungs were sampled? How many samples per lungs were taken? The distribution of the Sars-Cov-2 virus within the lungs may not be homogenous and some lung regions may be more colonized than others. Maybe, to pool several lung samples of the same animal together and make a lung homogenate from which DNA could be extracted, could increase the chance to detect the virus.

Figure 1. The figure present images of the lungs of T. truncatus. Did the authors try to detect Sars-Cov-2 by IHC in lungs of S. coeruleoalba or other aquatic mammalian species?

Paragraph entitled: “Effects of age and gender on ACE2 expression in cetacean lung tissues”: the term captivity should be added in the title since the authors also addressed whether captivity influenced the ACE2 expression

Table 2. The p-values are above 0.05, thus the Odds ratios cannot be interpreted. More samples need to be analysed to draw a definite conclusion. Do the authors plan to contact other governmental surveillance organization from other countries to get a broader overview at the Mediterranean sea scale? Along that line, there is no stranding sites on the Adriatic coast of Italy (Fig. 6). Are there any reasons for that?

There is a discrepancy between the number mentioned in Table 2 (p151, Odds ratio for Origin category = 4.8) and line 141 in the text (3.8).

Also, IHC is a qualitative or semi-quantitative approach to get information on where the protein is expressed and by which cell type ; but IHC is not really quantitative : would it be possible to measure the level of ACE2 expression by RT-qPCR?

How the other have scored the samples between high and weak expressors? This should be clarified in the Materials and Methods.

Figure 4. The column displaying the Dapi-only stain can be removed to increase the size of each image or to add an inset displaying higher magnification in order better visualize the labeled cells. Scale bar should be indicated.

In samples 133 and 343 (Fig. 4), where no colocalization was found (specimen 133 and 343), other cell marker should be tested to identify the cells expressing ACE2. Negative controls should be included to assess the background / non-specific staining. In the legend of Figure 4, the origin of the samples (lungs from T. truncatus) should be precised.

Western blot analysis (Figure 5): Positive controls of ACE2 and CD68 from human origin should be included in this figure. Although a band is highlighted by the anti-ACE2 and anti-CD68 antibodies on S. coeruleoalba brain protein extract (Fig. 5A-B), no band of the expected size seems to be recognized on the T. truncatus samples (Fig. 5C-D). Comment on these results would be appreciated as the IF colocalization results are presented for T. truncatus only (Fig. 4).

Figure 5. Why did the authors use brain samples (and not lung samples) to confirm the anti-CD68 and anti-ACE2 specificity by Western blot?

Table 3. Instead of having a Y/N answer in the IF colocalization column, authors may want to use a more objective way of quantifying the colocalization between these 2 markers by using the Pearson colocalization coefficient or the Manders’ overlap coefficient. The authors should precise on which basis the 4 samples presented in table 3 were selected.

Discussion

Line 210-211: As a closed sea, the Mediterranean sea may be more prone to concentrate water-borne pathogens. Has the Sars-Cov-2 pollution of Mediterranean sea been assessed at all? This point should be added in the introduction or the discussion.

Lines 309-313: ACE2 sequence homology between dolphin and human is a first indicator of the susceptibility of dolphins to Sars-Cov-2. Another way of tackling this issue would be to express the dolphin ACE2 in a cell culture model and try to infect these cells with Sars-Cov2.

The way of contamination of the dolphins by the virus is not discussed: inhaled contaminated droplets, nebulization, other ways?. This point should be addressed.

Materials and Methods

Samples have been collected from 2020 until 2022. The dates should be modified (line 325)

The sample n°43 is indicated to be T. truncatus in table 4 but it is identified as S. coeruleoalba in table 5. For better clarity and to avoid this kind of mistake, the tables 4 and 5 should be fused. The last column of table 5 can be removed as all the specimen tested in IHC were sampled from the lungs. This could be indicated in the legend of the table.

Minor English spelling errors:

- line 30: “emphasis” instead of “enphasis”

- line 38: “thirty-two” instead of “thirtytwo”

Results:

- line 130: “truncates” instead of “truncatus

- line 123 (legend of Figure 1): “T. trucatus” instead of “T. tuncatus”.

Round 2

Reviewer 2 Report

The reviewer is very satisfied with the answers and modifications proposed by the authors which have improved the quality of the manuscript.

Just a few things remaining on the second version of the submitted manuscript

Point 10. The figure 4 is very much improved with the higher magnification of the merged images. Although the reviewer adviced to remove the Dapi-only image, she thinks that keeping the dapi in the merged image together with the ACE-2 and CD68 labeling would be very much informative. Would it be possible to add the Dapi staining back in the merged image, please ?

Point 11. Reference to the supplementary figure showing the negative control for anti-CD68 and anti-ACE2 immunofluorescence should be make in the text of material and methods (line 429-430) as it has been added in the results text.

Point 12. Maybe an asterisk should be added to point out the specific band especially for the ACE2 western blot (Figure 5, A and C).

Point 14. Mention of Y/N should be removed in the table 3 legend (line 219).
